# Combined Photothermal Therapy and *Lycium barbarum* Polysaccharide for Topical Administration to Improve the Efficacy of Doxorubicin in the Treatment of Breast Cancer

**DOI:** 10.3390/pharmaceutics14122677

**Published:** 2022-12-01

**Authors:** Lina Sun, Cuiling Zuo, Xinxin Liu, Yifei Guo, Xiangtao Wang, Zhengqi Dong, Meihua Han

**Affiliations:** 1Institute of Medicinal Plant Development, Chinese Academy of Medical Sciences & Peking Union Medical College, Beijing 100193, China; 2Research Center of Pharmaceutical Engineering Technology, Harbin University of Commerce, Harbin 150076, China

**Keywords:** breast cancer, doxorubicin, *Lycium barbarum* polysaccharide, topical administration, photothermal therapy

## Abstract

In order to improve the efficacy of doxorubicin in the treatment of breast cancer, we constructed a drug delivery system combined with local administration of *Lycium barbarum* polysaccharides (LBP) and photothermal-material polypyrrole nanoparticles (PPY NPs). In vitro cytotoxicity experiments showed that the inhibitory effect of DOX + LBP + PPY NPs on 4T1 cells under NIR (near infrared) laser was eight times that of DOX at the same concentration (64% vs. 8%). In vivo antitumor experiments showed that the tumor inhibition rate of LBP + DOX + PPY NPs + NIR reached 87.86%. The results of the H&E staining and biochemical assays showed that the systemic toxicity of LBP + DOX + PPY NPs + NIR group was reduced, and liver damage was significantly lower in the combined topical administration group (ALT 54 ± 14.44 vs. 28 ± 3.56; AST 158 ± 16.39 vs. 111 ± 20.85) (*p* < 0.05). The results of the Elisa assay showed that LBP + DOX + PPY NPs + NIR can enhance efficacy and reduce toxicity (IL-10, IFN-γ, TNF-α, IgA, ROS). In conclusion, LBP + DOX + PPY NPs combined with photothermal therapy can improve the therapeutic effect of DOX on breast cancer and reduce its toxic side effects.

## 1. Introduction

According to the latest reports, breast cancer has gradually become one of the five most deadly cancers and the second most common cause of death among women worldwide [1,2,3,4]. At the same time, if malignant lesions occur in the catheter behind the areola in men, there is also a risk of breast cancer [5]. Among the existing treatments for breast cancer, including radiotherapy and chemotherapy, hormonal therapies, ligands, mono-clonal antibodies, targeted nanomedicines, and new approaches to individualized therapy [6], chemotherapy is the most commonly used and effective. However, chemotherapy can also cause various side effects, including cardiotoxicity, liver toxicity, and kidney toxicity [7]. Among chemotherapeutic agents, anthracyclines are valued for their powerful antitumor effects. The Early Breast Cancer Trials Collaborative Group (EBCTCG) meta-analysis conducted in 2012 showed that patients using anthracyclines (plus paclitaxel) had a one-third lower risk of breast cancer recurrence and 20–25% lower breast cancer mortality over 10 years compared to no chemotherapy [8]. Several countries, including the United States and France, have conducted clinical follow-up studies to obtain clinical data [9], and these studies have shown that anthracyclines can significantly reduce breast cancer recurrence rates and reduce mortality [10]. Doxorubicin (DOX) is a type of DNA topoisomerase II inhibitor that belongs to the anthracycline class of drugs with broad-spectrum antitumor activity and is one of the most commonly used chemotherapy drugs in the treatment of breast cancer [11]. Although it has good antitumor activity, the toxic side effects of doxorubicin on the nervous system, heart, liver, and kidneys, especially cardiotoxicity, have affected its further clinical development and application [12]. In order to reduce the toxicity of doxorubicin, researchers have used low-dose, long-term, and continuous administration, co-administration with toxicity-reducing drugs or protective agents, and liposome-encapsulation technology or drug carriers to prepare nano-targeted drugs that alter the in vivo distribution of the drug, thereby reducing the level of the drug at non-tumor sites to achieve toxicity reduction and increase its efficacy [13,14,15,16]. By contrast, local administration is a more effective way to administer the existing doxorubicin treatments for breast cancer, which can effectively improve drug distribution at the tumor site while reducing the drug’s accumulation at non-tumor sites and can reduce the administered dose to a certain extent, thus achieving better treatment outcomes [17,18]. Therefore, in the present investigation, we explored local administration methods to improve the efficacy of doxorubicin in breast cancer treatment, aiming to improve the therapeutic effects and reduce the toxic side effects of doxorubicin through multi-faceted synergistic treatment.

*Lycium barbarum* is a traditional Chinese medicine with the effect of nourishing Yin, tonifying the kidney, protecting the liver, and brightening the eyes. *Lycium barbarum* polysaccharide (LBP) is a class of water-soluble sugar-coupled active substance isolated and extracted from *Lycium barbarum* with a molecular weight of 10–2300 kDa [19]. The reported biological activities of LBP include anti-aging, antioxidant, metabolism-promoting, immunomodulatory, anti-cancer, and neuroprotective effects [20]. In terms of antitumor activity, it has been demonstrated that the main active component of LBP (molecular weight 40–350 kDa) can inhibit the growth of H22 cells in vitro, induce apoptosis, disturb mitochondrial-membrane potential, and cause S-phase block, and it has no significant toxicity to mice in vivo [21]. It also attenuates immunosuppression and maintains antitumor immune responses in mice; in addition, systemic and local immune responses of H22 tumor-bearing mice have been induced [22]. Some experimental studies have demonstrated that the combination of LBP and doxorubicin has a very good effect on reducing toxicity and enhancing efficacy [23,24,25,26,27,28]. LBP can also inhibit the proliferation of the lung cancer cell line A549, BIU87, and Renca kidney cancer cells, and its immunomodulatory and hepatoprotective effects strongly suggest that LBP can enhance the efficacy of anti-cancer drugs in immunogenic tumors and reduce side effects [29,30,31,32,33,34,35]. In order to further enhance the antitumor effect, combined with photothermal therapy, which has few side effects, is non-invasive, and has excellent local treatment results [36,37,38,39,40,41], polypyrrole (PPY), which has good biocompatibility and high photothermal conversion efficiency, was selected as the photothermal conversion material for multifaceted combined synergistic treatment in order to improve the effectiveness of the local administration of doxorubicin for the treatment of breast cancer [42,43,44,45,46,47,48].

In this study, temperature-responsive nanoparticles of highly efficient near-infrared (NIR) conversion polymer polypyrrole (PPY) were prepared, and the prepared PPY NPs were co-administered intratumorally with DOX and LBP to effectively reduce the toxic side effects of doxorubicin, while synergistically improving the antitumor effect of DOX. The aim was to explore the role of the traditional Chinese medicine component LBP combined with photothermal therapy in improving the anti-breast cancer effect of DOX.

## 2. Materials and Methods

### 2.1. Materials

Doxorubicin (DOX) purchased from Anhui Zesheng Technology Co., Ltd. (Anqing, China); *Lycium barbarum* polysaccharide (LBP) purchased from Shanghai Jinsui Biotechnology Co., Ltd., China; ferric chloride hexahydrate and pyrrole purchased from Beijing Bailingway Technology Co.; AL204 analytical balance purchased from Mettler-Toledo Instruments Shanghai Co.; IKA-RCTB asicsx thermostatic magnetic stirrer purchased from IKA, Germany; Zetasizer Nano ZS particle = size analyzer purchased from Malvern Instruments, U.S.A.; KQ3200DB CNC ultrasonic instrument from Kunshan Ltd.; PURELABC lassic integrated water purifier was provided by ELGA, UK; circulating water multi-purpose vacuum pump purchased from Henan Keda Machinery Equipment Co. Ltd., Wuhan, China; 808-nanometer infrared semiconductor laser (model: MW-GX-808) purchased from Changchun Laishi Optoelectronics Co., LTD; pathology slicer provided by Leica Instruments Ltd., Shanghai, China; tissue spreader purchased from Kedi Instruments Ltd., Jinhua, Zhejiang Province, China; oven purchased from Tianjin Laibori Instruments Co. 

### 2.2. Cell Lines and Animals

The 4T1 mouse cell line was provided by the Cell Center of Peking Union Medical College. Female Balb/c mice (6–8 weeks old, 20 ± 2 g) were provided by Beijing Vidahe Laboratory Animal Technology Co. 

All animal experiments were performed in accordance with the animal experimental and ethical standards prescribed by the Chinese Institute of Medicinal Plants (Beijing, China). Cells at 37 °C and 5% CO_2_ (Sanyo, Osaka, Japan) were cultured in RPMI-1640 medium (Hy Clone, Logan City, UT, USA) containing 10% FBS, 100 U/mL penicillin and streptomycin (Gibco, St Louis, MO, USA). Experimental animals were acclimated to a standard diet at 25 °C for 1 week.

### 2.3. Preparation and Characterization of PPY NPs

#### 2.3.1. Preparation of PPY NPs

The polyvinyl alcohol solid dissolved in water was weighed to prepare a concentration of 37 mg/mL of polyvinyl alcohol solution, after which the ferric chloride hexahydrate solid added to the prepared polyvinyl alcohol solution was weighed and stirred at 75 °C for 1 h; the resulting mixed solution was then transferred to a 4 °C ice–water bath, after which pyrrole monomer solution was added, and stirred in the ice–water bath for 4 h to obtain the polypyrrole solution after 13,000 rpm centrifugation for 6 min. The solution was washed 3 times with water and then lyophilized. The mass–volume ratio of ferric chloride hexahydrate to pyrrole monomer solution was 9:1 [49,50].

#### 2.3.2. Characterization of PPY NPs

##### Particle Size and Zeta Potential

The size, zeta potential, and polydispersity indices (PDIs) of PPY NPs were measured using Dynamic Light Scattering (DLS, Zeta Sizer Nano ZS, Malvern Instruments, UK) at room temperature. The PPY NP samples were measured three times in parallel. The morphology of PPY NPs was confirmed by photographing with a transmission electron microscope (HT7700, Hitachi, Tokyo, Japan).

##### Stability of PPY NPs in Physiological Medium

PPY NPs were mixed with 1.8% sodium chloride and 10% glucose solution of the same volume, mixed with two times the concentration of PBS at a volume ratio of 1:1, and incubated at 37 °C. Particle size and PDI were measured at certain time intervals, three times in parallel (n = 3).

#### 2.3.3. Investigation of Photothermal Conversion Performance

One milliliter of PPY NPs was added to the cuvette, and a temperature sensor was inserted. After preparation and stabilization, the solution temperature recorded was the solution temperature at 0 s. The solution was subsequently irradiated with 808 nm NIR at a fixed power of 3.5 W/cm^2^. The solution temperature was recorded every 30 s to adjust the concentration of nanoparticles and eventually determine the concentration of nanoparticles. 

### 2.4. In Vitro Cellular Uptake

The 4T1 cells in logarithmic growth phase were inoculated in 24-well plates with 5000 cells per well and incubated at 37 °C and 5% CO_2_ for 24 h. Next, a DOX (10 μg/mL) and LBP (5 mg/mL) solution was prepared using RPMI-1640 culture medium and 0.5 mL was added to each well; after 3 h, 0.5 mL of 4% paraformaldehyde was added to each well. After fixation for 20 min, the liquid was discarded, and 0.4 mL of DAPI solution was added to each well for staining for 10 min. The liquid was discarded, 0.3 mL of PBS was added, and the DOX uptake of 4T1 cells was observed using a fluorescence microscope imaging system. After each discarding of the liquid in the well plate, the plate was washed twice with PBS.

### 2.5. In Vitro Cytotoxicity Assay

The MTT assay was used to assess the in vitro cytotoxicity of DOX, LBP, PPY NPs, and DOX + LBP + PPY NPs combined with photothermal treatment groups. The cell culture conditions were 37 °C, 5% CO_2_. The 4T1 cells in logarithmic growth phase were inoculated into 96-well plates at 8000 cells/well and cultured under cell culture conditions for 24 h. Next, the drug was diluted with RPMI-1640 and free DOX solution at concentrations of 0.01, 0.05, 0.2, 0.5, 1, 2, 3, 4, and 5 μg/mL, free LBP solution at concentrations of 1.5, 5, 10, 50, 75, 100, 125, and 150 mg/mL, PPY NPs solution at concentrations of 0.1, 0.5, 1, 2.5, 5, 20, 40, and 50 mg/mL, and concentrations of 0.05, 0.1, 0.5, 1.5, and 3 mg/mL of PPY NPs + NIR (at 3.5 W/cm^2^ power and 808 nm NIR radiation for 180 s) for 150 μL per well (6 wells for each sample). The 96-well plates of each drug were used with RPMI-1640 culture solution as blank controls. After continuing to culture for 72 h, MTT solution (20 μL, 5 mg/mL) was added to each well to treat 4T1 cells and culture was continued for 4 h. Finally, 200 μL of DMSO was added to each well and the maximum absorbance was measured at 570 nm using an ELISA plate reader (Biotek, Winooski, VT). After obtaining the IC_50_ values for the four cases above, the respective experimental concentrations were selected to compare the inhibition rates of 4T1 cells at the same concentration alone with those of DOX + LBP, DOX + PPY NPs + NIR, and DOX + LBP + PPY NPs + NIR cells. The cell-inhibitory rate was calculated using Equation (1):Cell-inhibitory rate (%) = 1 − (ODe/ODc) × 100%(1)
where ODe and ODc are the average optical density of the experimental group and control group. 

The IC_50_ value (half-maximal inhibitory concentration) of cells was calculated using GraphPad Prism 6 software (GraphPad software, Inc., La Jolla, CA, USA).

### 2.6. In Vivo Tissue-Distribution Study

The 4T1 cells at a concentration of 1.0 × 10^7^ cells/mL were injected subcutaneously with 0.2 mL in the right axilla of each female Balb/c mouse. When the tumor volume reached 100 mm^3^, the mice were randomly divided into 3 groups (3 animals per group). The administration dose was 2 mg/kg DOX, 62.5 mg/kg LBP, and 5 mg/kg PPY NPs. The experimental groupings are shown in Table 1. After 24 h, mice were euthanized and the hearts, livers, spleens, lungs, kidneys, and tumors were collected and then fluorescently irradiated with IVIS Living Image (Caliper Life Sciences, Hopkinton, MA, USA) to obtain the fluorescence distribution of each tissue.

### 2.7. In Vivo Antitumor Study

To investigate the vivo antitumor effect of intratumoral local delivery systems of DOX, DOX + LBP, and PPY NPs in combination with photothermal therapy and chemotherapy in tumor-bearing (4T1) Balb/c mouse model, we selected mice with similar tumor volumes (about 100 mm^3^) and randomly divided them into 8 groups (6 animals per group). The administration dose was 2 mg/kg DOX, 62.5 mg/kg LBP, and 5 mg/kg PPY NPs. The treatment status of the subgroups is shown in Table 2. All of the NIR groups were irradiated with 808 nm NIR at a power of 3.5 W/cm^2^ for 180 s after intratumoral injection. The dosing cycle for all the groups was 12 days. The tail vein-injection group was administered a dose of 0.2 mL per mouse once every two days, and the intratumoral local dosing group was administered a dose of 0.1 mL per mouse only once on day 0 and twice on day 4. The abnormalities in and deaths of mice were observed every day.

Mice were weighed and their width a and length b were measured to calculate the tumor volume using the formula: V = (a × b^2^)/2. Tumor volumes and mouse weights were recorded every 2 days during the experiment. At the end of treatment, mice were euthanized and dissected. We dissected the complete tumor, liver, and spleen of each mouse. We obtained the tumor, heart, liver, spleen, lung, kidney, and thymus of each mouse by dissection. Each organ obtained was weighed to calculate CIR (cardiac index rate), LIR (liver index rate), SIR (splenic index rate), LuIR (lung index rate), RIR (renal index rate), and TIR (thymus index rate). Tumor survival rate (TSR) can be calculated according to the following equation (Equation (2)):TSR (%) = Wt/Wn(2)

The CIR, LIR, SIR, LuIR, RIR, and TIR can be calculated as below (Equation (3)):CIR (%) = WC/Wm, LIR (%) = WL/Wm, SIR (%) = WS/Wm, LuIR (%) = WLu/Wm, RIR (%) = WR/Wm, TIR (%) = WT/Wm(3)
where Wt means the tumor weight in the administration group, except the saline group, and Wn means the tumor weight in the negative control group. WC, WL, WS, WLu, WR, and WT mean the cardiac weight, liver weight, spleen weight, lung weight, renal weight, and thymus weight of the test group, respectively. Wm is the mouse weight in the group.

### 2.8. H&E Staining

To test the in vivo toxicity in mice with different administration methods, we used the same protocol as in vivo antitumor studies for Balb/c mice, where DOX solution alone was administered in tail veins, while DOX + LBP solution and DOX + LBP + PPY NPs were administered intratumorally and locally, and the same conditions were used in the combined group for photothermal treatment. At 24 h after the last administration, all mice were euthanized. Tumors and major organs, such as heart, liver, spleen, lung, and kidney, were obtained by autopsy, fixed in 10% formalin for 2 d, paraffin-embedded, and cut into 10-supplier sections. Each tissue section was stained with hematoxylin-eosin (H&E) using an ortho-optical microscope (Nikon Eclipse E100, NIKON DS-U3, Japan).

### 2.9. Serological Analysis

Serum samples of saline, DOX (iv), LBP + DOX (R), and LBP + DOX + PPY (R + NIR) were collected in centrifuge tubes and centrifuged at 3000 rpm for 5 min at 4 °C. The serum levels of alanine aminotransferase (ALT) and aspartate transaminase (AST) were measured using a fully automated biochemical analyzer (Chemray 240, Shenzhen Redu Life Sciences, China). At the same time, serum samples of DOX (iv), LBP + DOX (iv), DOX + PPY (R + NIR), and LBP + DOX + PPY (R + NIR) were prepared using the same serum preparation method, and serum levels of inflammatory cytokines, including IL-10, IFN-γ, TNF-α, IgA, and ROS, were determined using ELISA kits (Beijing Dongge Boye Biotechnology Co., Ltd., Dogesce Beijing, China).

### 2.10. Statistical Analysis

Statistical analysis was conducted via one-way ANOVA (nonparametric) with GraphPad Software 6 (GraphPad Software Inc., La Jolla, CA, USA) to test the statistical significance between the experimental groups.

## 3. Results

### 3.1. Preparation and Characterization of PPY NPs 

The particle sizes of nanoparticles are favorable for pharmacokinetics and cellular uptake in vivo [51]. As shown in Figure 1A,B, the particle size of the PPY NPs was 229.3 ± 2.696 nm, and the PPY NPs were round and spherical under transmission electron microscopy. It can be seen from the particle size and PDI values that, since the PPY NPs had smaller particle size and narrower particle size distribution, they had better dispersion in water. Therefore, the prepared PPY NPs improved not only the solubility, but also the efficacy of the drug.

### 3.2. Stability of PPY NPs in Various Physiological Media 

PPY NPs were incubated in different physiological media (saline, glucose, PBS) at 37 °C. Samples were taken at fixed-interval time points during the incubation process for testing. The PPY NPs displayed no turbidity or precipitation within 12 h, and there was no significant increase in particle size and PDI, which indicated that the PPY NPs could exist stably in various physiological media (Figure 1C,D).

### 3.3. Photothermal Conversion Performance of PPY NPs

We investigated the temperature changes in different concentrations of PPY NPs under NIR irradiation to study the photothermal conversion performance of PPY NPs and to determine the optimal conditions of the laser irradiation used for the experiments. As shown in Figure 2, the photothermal conversion performance of the PPY NPs increased as the concentrations increased, and we found that the PPY NPs of 0.5 mg/mL can increase the temperature by about 7 °C after being irradiated at a power of 3.5 W/cm^2^ and 808-nanometer NIR for 180 s, which can make the local temperature of the animal rise to more than 42 °C under this condition, so as to facilitate further experimental research.

### 3.4. In Vitro Cellular Uptake

To further investigate the therapeutic effect of the combined application of DOX and LBP, we performed cell uptake experiments on the 4T1 cells with DOX and DOX combined with LBP and set up a combination of DOX with the LBP experimental control group under constant DOX concentration. As demonstrated in Figure 3A, the cell uptake results of both groups showed DOX uptake into the cells, and compared with the DOX-alone administration group, the DOX + LBP group showed stronger DOX fluorescence intensity, which may have been related to the fact that the LBP promoted the DOX uptake into the cells [52,53].

### 3.5. In Vitro Cytotoxicity Assay

IC_50_ is an index used to assess the in vitro antitumor effect, which is the concentration of a drug that causes apoptosis in 50% of tumor cells. The cytotoxic effect of the drug on the 4T1 cells was detected by tetramethylazole salt colorimetric assay (MTT), after which we evaluated the toxicity of the combination treatment on the 4T1 cells. The IC_50_ values of the DOX solution and LBP solution on the 4T1 cells were 1.219 μg/mL and 8.401 mg/mL, respectively (Appendix A), and the IC_50_ value of the PPY NPs + NIR on the 4T1 cells was 1.127 mg/mL (Appendix A), while different concentrations of PPY NP solution had no significant inhibitory effect on the cells (Appendix A), indicating that PPY NPs have good biocompatibility. Since the IC_50_ value of the DOX on the 4T1 cells was 1.219 μg/mL, the concentration of DOX was chosen to be 0.8 μg/mL in the cytotoxicity experiment for the control comparison of cell inhibition, making the cell survival rate of the DOX-alone group greater than 50%; similarly, the concentration of LBP was chosen to be 5 mg/mL and the concentration of PPY NPs was chosen to be 0.5 mg/mL. By choosing drug concentrations lower than the IC_50_ value concentration for the control test, not only were the dose and toxicity of the DOX reduced, but the difference in cytotoxicity between the alone-administration group and the combination group was compared. The experimental results showed that the cell survival rate of the DOX + LBP + PPY NPs combined with the photothermal treatment was only 8% (8% vs. 64%) compared with the DOX-alone group (Figure 3B), showing good antitumor and synergistic effects.

### 3.6. Drug Distribution Studies In Vivo

Twenty-four hours after the last dose, all the mice were euthanized and dissected, and the obtained tumors and individual organs were imaged in vitro to study the distribution in vivo. As shown in Figure 4A, fluorescence was predominantly distributed in the tumors both when the DOX + LBP solution was injected intravenously and when it was injected intratumorally. In addition, the fluorescence in the DOX + LBP combined with the PPY NPs photothermal treatment group was more uniformly distributed at the tumor site and further increased compared with the intravenous injection and the group without photothermal treatment, confirming that the NIR laser can increase the accumulation of the drug at the tumor site. This may be due to the fact that irradiation with an 808-nanometer NIR laser increases the intratumoral temperature, which facilitates drug penetration and release at the tumor site and drug endocytosis in cells [54,55].

### 3.7. In Vivo Antitumor Research

The dosing schedule is shown in the Appendix A. Two days after the laser irradiation, the tumors of the mice showed significant crusting. This indicates that this treatment kills tumor cells. However, tumor-site crust was not convenient for tumor injection and, considering that the distribution of the drug at the tumor site was greater after local intratumoral administration, the frequency of local intratumoral administration was set to a total of 2 injections on day 0 and day 4. The frequency of intravenous drug administration was once every 2 days for a total of six times.

In Figure 4B, it can be seen that the tumor volume in the saline group increased rapidly, as well as in the PPY (R + NIR) group, but that the tumor volume was slightly smaller than that in the saline group, which was consistent with the results of the in vitro cytotoxicity experiments in the PPY (R + NIR) group. Compared with the DOX (iv) group, the tumor inhibition rate of DOX (R) was significantly higher, which showed the antitumor advantage of local intratumoral administration. With both intravenous and intratumoral injection, the tumor volume in the DOX group was larger than in the DOX + LBP group, indicating that LBP can synergize with DOX to enhance antitumor effects [23]. The tumor volume in the DOX (R) group was larger than that in the DOX + PPY (R + NIR) group, and the tumors in the DOX + LBP (R) group were larger than those in the DOX + LBP + PPY (R + NIR) group, indicating that biocompatible photothermal conversion material PPY NPs can improve the absorption efficiency of NIR by tumor tissues, which also shows an excellent effect after combined photothermal treatment. The tumor volume in the LBP + DOX + PPY NPs + NIR group was significantly reduced compared with the saline group, with statistical differences (Figure 4C) (*p* < 0.05). The highest tumor inhibition rate was observed in the LBP + DOX + PPY NPs + NIR solution group, which was significantly better than the other groups (87.86%) (Figure 4D). It was demonstrated that DOX combined with LBP, and PPY NPs with simultaneous photothermal treatment significantly improved the antitumor efficacy.

Mouse bodyweight reflects the systemic toxicity of a drug during administration and is one of the evaluation indicators of drug safety. The bodyweights of the mice, which were recorded every other day during the 12-day dosing cycle, are shown in the Appendix A. It can be seen that, except for the DOX (iv) group and the DOX + LBP (iv) group, the bodyweights of the mice in all the groups increased slightly compared with before the drug administration. This indicates that intratumoral administration combined with photothermal treatment can significantly reduce the systemic toxicity of DOX. The bodyweights of the DOX + LBP (iv) group were significantly lower than those of the saline group and the DOX + LBP + PPY (R + NIR) group. The difference was statistically significant (*p* < 0.05), but no mice died in any of the groups during the experimental period. Compared with intravenous DOX alone, the local intratumoral administration of DOX, LBP, and PPY NPs and the combination with photothermal therapy reduced the systemic distribution of the drug, thus reducing the systemic toxicity.

The organ index (CIR, LIR, SIR, LuIR, RIR, TIR) can be used to evaluate the degree of drug damage to various organs in the body. There was no significant difference in the organ index between each group and the normal saline group. The results showed that the intratumoral injection combined with photothermal therapy caused low toxicity in the internal organs (Appendix A).

### 3.8. H&E Staining

Twenty-four hours after the last intravenous administration, the mice were euthanized, and the organs and tumors of each group were examined anatomically and histopathologically to further evaluate the magnitude of the toxic effects and the tumor-suppressive effect. After the H&E staining of the normal tissues or cells, the nuclei were stained blue–purple by alkaline hematoxylin, and the cytoplasm and extracellular matrix were stained pink by acid eosin. From the H&E staining shown in Figure 5, it can be seen that no obvious damage took place in the organs of the saline group, and most of the tumor cells were round and irregularly arranged, with more chromatin and good growth. Compared with the saline group, the organ damage to the hearts and livers in the DOX (iv) group was relatively large, and the morphologies of the tumor cells changed, while the organ damage in the LBP + DOX (R) group was relatively reduced, and the tumor tissue damage was increased with fragmentation, showing that the addition of LBP reduced the damage to the organs by the DOX and increased the antitumor ability at the same time, that the damage to the tumor tissues was further increased, and that the irregularity and fragmentation of the tumor cell traits were significantly enhanced by the direct topical administration to the tumor tissues. From the figure, it is easy to find that, compared with the LBP + DOX (R) group, no obvious organ damage was seen in the LBP + DOX + PPY (R + NIR) group, but that the number of tumor cells was significantly reduced and that the tumor cells showed more irregular shapes, vacuolization, cytoplasmic lysis, nuclear consolidation, fragmentation, and even disappearance of the solution, as well as a small number of intact cell outlines and nuclei. Therefore, the co-administration group significantly reduced the organ damage caused by the DOX, while the tumor necrosis reflected that the co-administration had superior antitumor ability. 

### 3.9. Serological Analysis

To further illustrate the potentiation and toxicity-reduction effects of the DOX combined with the LBP photothermal treatment, we first examined the serum indices of the liver function, including alanine aminotransferase (ALT) and aspartate aminotransferase (AST), in the experimental mice. We found that the serum ALT and AST levels were significantly higher in the DOX (R) compared with the saline control group, while the serum ALT and AST were significantly lower in the LBP + DOX + PPY photothermal treatment group compared with the DOX-alone treated group, and the difference between the two groups was statistically significant (Figure 6A,B) (ALT 54 ± 14.44 vs. 28 ± 3.56; AST 158 ± 16.39 vs. 111 ± 20.85) (*p* < 0.05). This indicates that the addition of LBP combined with photothermal treatment can significantly reduce the liver injuries caused by DOX [56,57].

Since LBP is known to induce anti-inflammatory and immune responses, we quantified the levels of the secreted cytokines, including IL-10, IFN-γ, TNF-α, IgA, and ROS, in the sera of the mice with LBP added to their corresponding control groups. IL-10, IgA and ROS were significantly decreased (Figure 6C–E), while the IFN-γ and TNF-α levels were significantly increased (Figure 6F,G), indicating that the addition of LBP inhibited the secretion of IL-10, IgA, and ROS in the mice. Recently, IgA began to be considered as a key biomarker of immunosuppressive B cells; IgA can encourage the formation of tumor-immunosuppressive environments and reduce the antigen delivery and phagocytosis of tumor cells, and the antibodies produced by IgA can form immune complexes with tumor or non-tumor antigens, interact with immunosuppressive cells, including MDSCs cells, increase inflammatory responses, and encourage immune escape. Thus, LBP causes the reduction of IgA levels, reducing the suppressive effect of antitumor immunity [58]. ROS has been shown to promote cell proliferation, activate pro-tumor signaling, and facilitate tumor resistance, and LBP reduces the level of ROS, thereby exerting an antitumor effect [59,60]. As an immunosuppressive factor, IL-10 can inhibit the expression of MHC-II to reduce antigen presentation, thus inducing immunosuppression to promote tumor escape; therefore, the reduction in IL-10 levels plays an immune activation and anti-cancer role to some extent. At the same time, the reduction in IL-10 secretion can alleviate the inhibition of IFN-γ and TNF-α secretion [61,62,63], IFN-γ can reduce the number of tumor stem cells, and the increase in IFN-γ level responds to its anti-proliferative, apoptosis-promoting, and antitumor effects [64,65,66], while the increase in TNF-α content indicates the increase in mono-nuclear macrophages in tumor cells [67], which means that LBP increases the antitumor ability of DOX.

## 4. Conclusions

In conclusion, DOX + LBP + PPY NPs were combined with photothermal treatment for the local delivery of breast cancer treatment. This combination of the known antitumor, anti-inflammatory immune and hepatoprotective effects of LBP with biocompatible photosensitive material PPY NPs for photothermal treatment to reduce the dose of DOX and the systemic distribution of the drug can not only substantially enhance the antitumor effect of DOX, but can also reduce systemic toxicity during DOX treatment and improve the anti-inflammatory immune effect. In conclusion, the proposed DOX + LBP + PPY NPs combined with photothermal treatment for local drug delivery for breast cancer treatment is a promising way to combine herbal ingredients with chemotherapeutic drugs and photothermal therapy to achieve potentiation and toxicity reduction, which can be used for the further clinical treatment of cancer.

## Figures and Tables

**Figure 1 pharmaceutics-14-02677-f001:**
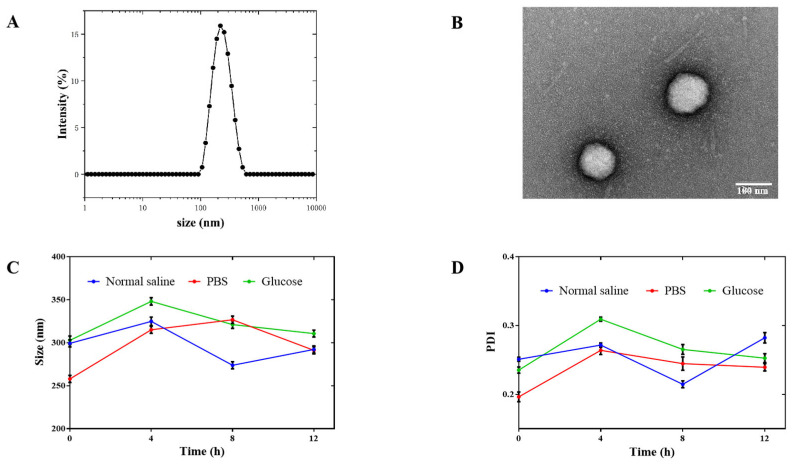
Particle size distribution curve of PPY NPs (**A**); TEM images of PPY NPs (**B**) (Scale bar 100 nm); the size (**C**) and PDI diagrams (**D**) for the stability of PPY NPs in physiological media (n = 3).

**Figure 2 pharmaceutics-14-02677-f002:**
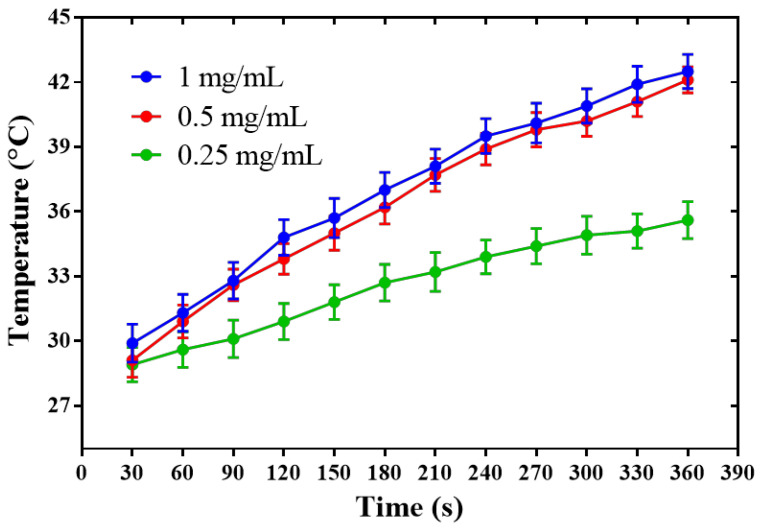
Photothermal conversion performance of PPY NPs with different concentrations.

**Figure 3 pharmaceutics-14-02677-f003:**
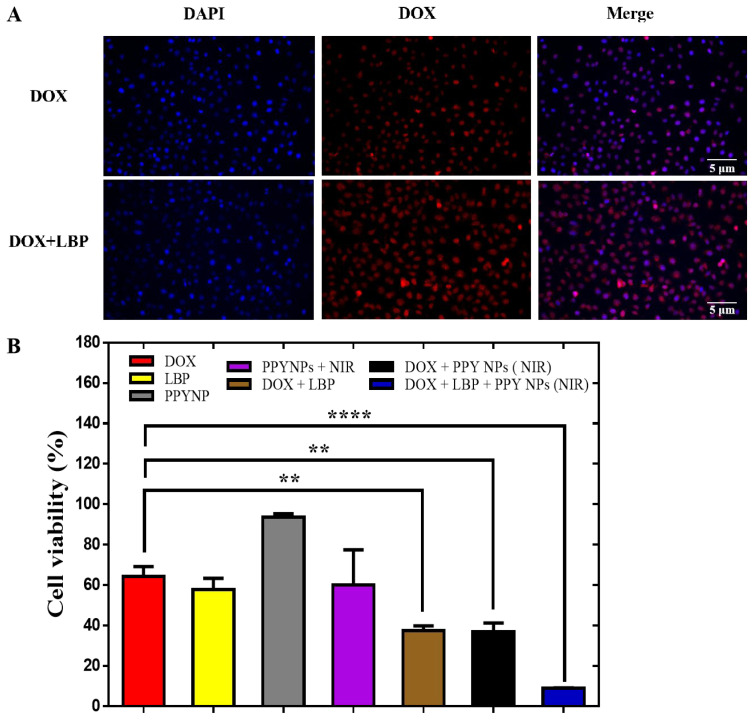
Fluorescence images of 4T1 cells after incubation with different drugs for 4h (**A**); in vitro cytotoxicity studies of DOX + LBP combined with PPY NPs (+NIR) against 4T1 cells at 72 h (n = 5) (**B**). ** *p* < 0.01, **** *p* < 0.0001.

**Figure 4 pharmaceutics-14-02677-f004:**
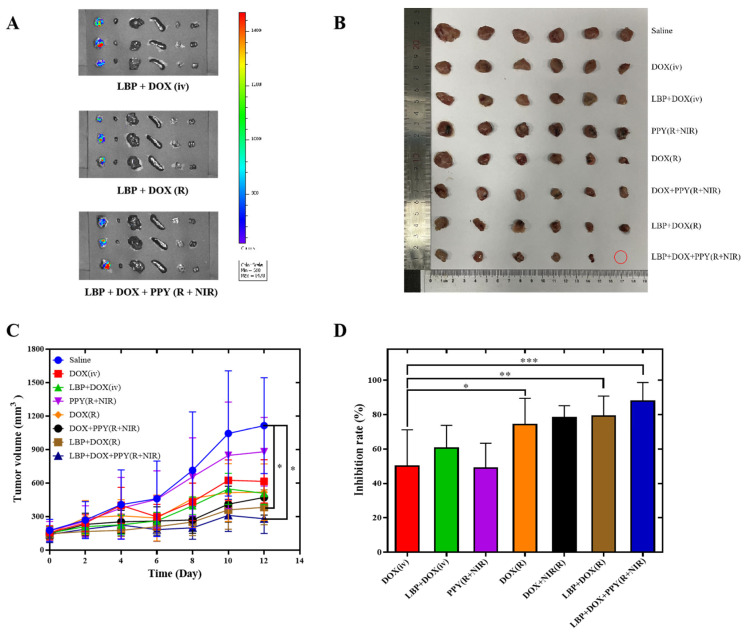
Distribution of DOX and LBP with different routes of administration and delivery forms in different tissues of 4T1 tumor-bearing mice. From left to right in the picture: tumors, hearts, livers, spleens, lungs, and kidneys (n = 3) (**A**); anatomical diagram of all groups of solid tumors at the end of the experiment (**B**); tumor growth curves (**C**) (mean ± SD, n = 6); tumor inhibition rate (**D**) (mean ± SD, n = 6). * *p* < 0.05, ** *p* < 0.01, *** *p* < 0.001.

**Figure 5 pharmaceutics-14-02677-f005:**
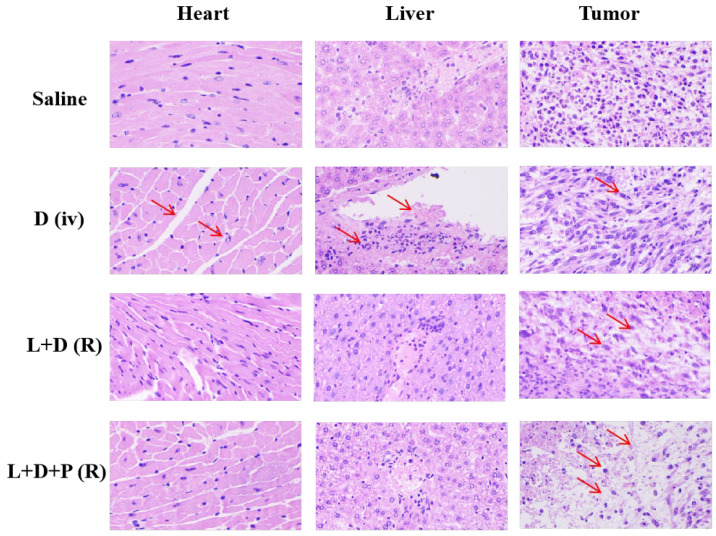
H&E-staining images of heart, liver, and tumor slices (D (iv) refers to DOX (iv), L + D (R) refers to LBP + DOX (R), L + D + P (R) refers to LBP + DOX + PPY (R + NIR)). The place indicated by the red arrow is the site of injury or lesion.

**Figure 6 pharmaceutics-14-02677-f006:**
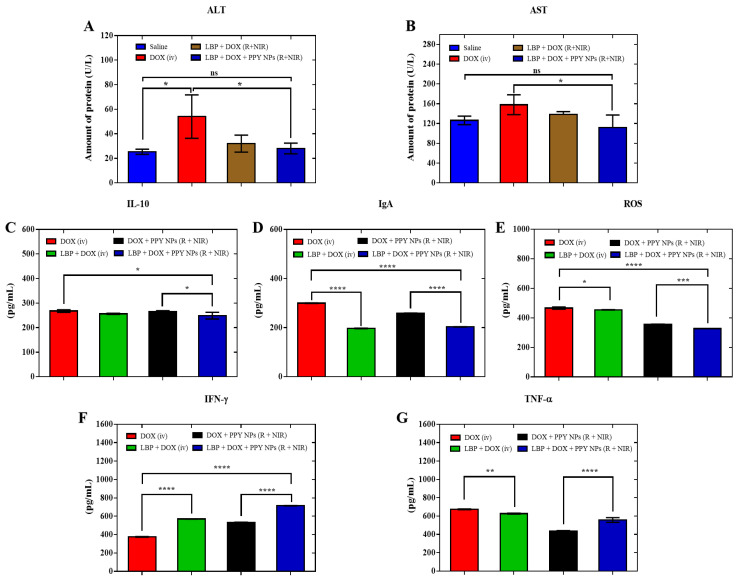
ALT (**A**) (n = 3; * *p* < 0.05); AST (**B**) (n = 3; * *p* < 0.05); ELISA of cytokines in mice sera injected with samples: IL-10 (**C**), IgA (**D**), ROS (**E**), IFN-γ (**F**), and TNF-α (**G**) (n = 4; * *p* < 0.05, ** *p* < 0.01, *** *p* < 0.001, **** *p* < 0.0001, ns, not statistically significant.). Results are presented as the mean ± SD.

**Table 1 pharmaceutics-14-02677-t001:** Experimental grouping of tissue distribution.

Group Name	Mode of Administration	Volume	Near-Infrared Processing Parameters
LBP + DOX (iv)	Tail vein injection	0.2 mL	-
LBP + DOX (R)	Intratumoral injection	0.1 mL	-
LBP + DOX + PPY (R + NIR)	Intratumoral injection	0.1 mL	808 nm, 3.5 W/cm^2^, 180 s

**Table 2 pharmaceutics-14-02677-t002:** In vivo experimental grouping.

Group Name	Mode of Administration	Volume	Near-Infrared Processing Parameters
Saline	Tail vein injection	0.2 mL	-
DOX (iv)	Tail vein injection	0.2 mL	-
LBP + DOX (iv)	Tail vein injection	0.2 mL	-
PPY NPs (R + NIR)	Intratumoral injection	0.1 mL	808 nm, 3.5 W/cm^2^, 180 s
DOX (R)	Intratumoral injection	0.1 mL	-
DOX + PPY NPs (R + NIR)	Intratumoral injection	0.1 mL	808 nm, 3.5 W/cm^2^, 180 s
LBP + DOX (R)	Intratumoral injection	0.1 mL	-
LBP + DOX + PPY NPs (R + NIR)	Intratumoral injection	0.1 mL	808 nm, 3.5 W/cm^2^, 180 s

## Data Availability

All the data obtained from the analyses reported in this article are available upon request.

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
