# Peer review of "Combined Photothermal Therapy and Lycium barbarum Polysaccharide for Topical Administration to Improve the Efficacy of Doxorubicin in the Treatment of Breast Cancer"

_pharmaceutics, 2022, doi:10.3390/pharmaceutics14122677_

Round 1
Reviewer 1 Report (New Reviewer)
In the submitted manuscript entitled "Combined photothermal therapy and Lycium barbarum poly- 2 saccharide for topical administration to improve the efficacy of 3 doxorubicin in the treatment of breast cancer ", the research article is focused in determined the importance as a possible treatment in breast cancer of Lycium barbarum saccaride combined with DOX. I have suggestions to improve the impact of researc. Comments, and suggestion to the author are the following.
My comments are as:
Minor comments:
It is necessary to put the names of the species in italics for example Lycium barbarum (line 71)
I consider that scheme 1 should be figure 7 to summarize the entire experimental design
Resolution of images must be improved.
Major comments
-The introduction does not mention anything about Lycium barbarum, in the part of previous treatment used in traditional chinese medicine.
-As demonstrated in Figure 3A, the cellular uptake results of both groups showed DOX uptake into cells, and compared to the DOX alone group, the DOX + LBP group showed higher DOX fluorescence intensity. strong, which may be related to the fact that LBP promoted DOX uptake in cells
-In my opinion, this result is the other way around, since the intensity is lower in the cells treated with DOX-LBP and the authors describe that they looked the same, even higher intensity in those with only DOX.
-Figure 4B seems to me to be the size of the tumors if it presents variation. The authors point out that both the intravenous and intratumoral routes of Dox show a similar decrease. It seems to me that I find that in intratumoral Dox it did decrease much more, just as when Dox+LBP is combined, the size of the tumor in intratumoral is much smaller. The comparisons in the results are little discussed since a significantly relevant reduction is not observed when only using PPY(R+NIR).
-If the intratumoral application of LBP+DOX+PPY(R+NIR) is more effective, apart from decreasing the concentration of the drug, making it more soluble and therefore having greater absorption by the cells, reducing the toxic effects of the drug and with it see an increase in the antitumor effect of DOX.
-It is necessary to mention what kind of statistical analysis was performed.
Author Response
Dear reviewer:
Thank you for your comments concerning our manuscript and offering us opportunity to improve the quality of our submitted manuscript entitled “Combined photothermal therapy and Lycium barbarum polysaccharide for topical administration to improve the efficacy of doxorubicin in the treatment of breast cancer”. Those comments are valuable and very helpful. We have read through comments carefully and have made corrections. Based on the instructions provided in your letter, we uploaded the file of the revised manuscript. We submit here the revised manuscript as well as a list of changes. The responses to the reviewer's comments are marked in red and presented following. Our point-to-point responses to the queries raised by the reviewers are listed, please see the attachment. We sincerely hope this manuscript will be finally acceptable to be published on Pharmaceutics: Nanomedicine and Nanotechnology (Polymer Nanoparticles for the Delivery of Anticancer Drugs). Thank you very much for all your help and looking forward to hearing from you soon. Sincerely wish you health, happiness and all the best!
Best regards
Sincerely yours
Meihua Han and Lina Sun

Reviewer 2 Report (New Reviewer)
This manuscript, “Combined photothermal therapy and Lycium barbarum poly- 2 saccharide for topical administration to improve the efficacy of 3 doxorubicin in the treatment of breast cancer” evaluated the local delivery method of Doxorubicin and constructed a drug-delivery system for the intratumoral local delivery of chemotherapy combined with photothermal therapy through the co-delivery of Lycium barbarum polysaccharide and Doxorubicin.
There are points that need to be checked.
Abstract
- The abstract exceeds the maximum of 200 words and needs to be shortened and revised following the style of structured abstracts.
Materials and methods
- Recommended to change hour/hours to h
- Line 136; change r/min to rpm
- All the equations should be prepared using the equation tool in MS word
- Line 203; 2.7 In 2.7 In vivo antitumor study to 2.7 In vivo Antitumor Study
- Line 235; 2.8 H&E stain2.8 H&E staining to 2.8 H&E Staining
Results
- The content of polymer polypyrrole (PPY) in PPY NPs should be analysed and discussed.
- The text size of several figures is too small; recommended to enlarge
- The error bar should be added in Figure I & II
- Use the same format of p<0.0xx
Author Response
Dear reviewer:
Thank you for your comments concerning our manuscript and offering us opportunity to improve the quality of our submitted manuscript entitled “Combined photothermal therapy and Lycium barbarum polysaccharide for topical administration to improve the efficacy of doxorubicin in the treatment of breast cancer”. Those comments are valuable and very helpful. We have read through comments carefully and have made corrections. Based on the instructions provided in your letter, we uploaded the file of the revised manuscript. We submit here the revised manuscript as well as a list of changes. The responses to the reviewer's comments are marked in red and presented following. Our point-to-point responses to the queries raised by the reviewers are listed, please see the attachment. We sincerely hope this manuscript will be finally acceptable to be published on Pharmaceutics: Nanomedicine and Nanotechnology (Polymer Nanoparticles for the Delivery of Anticancer Drugs). Thank you very much for all your help and looking forward to hearing from you soon. Sincerely wish you health, happiness and all the best!
Best regards
Sincerely yours
Meihua Han and Lina Sun

Reviewer 3 Report (New Reviewer)
The authors describe a straightforward combination of three materials, DOX, LBP and PPY-NPs together delivered to a tumor to seek to achieve synergistic antitumoral capabilities, and largely succeed. The approach demonstrates promise. The paper itself, while well written, could be improved by some attention to improving the figures. Specific comments follow:
Scheme 1 – the figure seems to be strangely cut off.
Line 94 – Here you discuss the number of peripheral blood lymphocytes induced by LBP. This is the first and only mention of it having this effect – I would either expand on it or take it out of the paper. As is, it is a peripheral (but interesting) assertion.
Methods and materials: You describe TSR, CIR, LIR, SIR, etc and then you do not, as far as I can tell, give results using these metrics. Please bring back in the results that use these metrics, for a stronger paper, or take out the metrics from methods and materials that you do not use in the results.
Line 263 – particle size is not 229.3 +/- 2.696 nm. Based on Figure 1A, particles ranged in size from about 100 – 400 nm and it looks to me like the peak is well under 200 nm. While this is a very reasonable size for PPY NPs injected, the difference in results in the text vs. Figures does not give readers confidence. Scale bar in Fig 1B is illegible, fix. Also fix your significant figure reporting here and throughout.
Figure 2. Your graph suggests irradiation of 0.5 mg/mL for 180 seconds will yield temp of 36C. Yet you state these are the optimal conditions to obtain irradiation above 42C. This does not make sense. I think what you mean to say is ΔT is 7C for 0.5mg/mL at 180 s, which will raise temp in animal above 42C, but I’m not sure. Please clarify.
Fig 3A, image is odd because DAPI so much weaker in the Dox+LBP cell group. Can you get a better picture, where DAPI is more similar in DOX and DOX+LBP? If not, can you quantify the effect, perhaps by counting # nuclei with uptake divided by all nuclei (looks to be near 1 for Dox+LBP and maybe 75% or worse for Dox alone). Again, your scale bar is unreadable, which is good, because it reads, when I blow it up, “some number of pixels”. Must read size in length (mm, cm, μm, not pixels).
Fig 3B. I would suggest you add significance bars. Also, I don’t understand how if IC50 of PPY NPs+NIR was 1.127 mg/mL and PPY NP alone had no toxicity, you see no difference in IC50between PPY NPs + NIR and PPY NPs alone. Rerun this experiment to make sure it is correct or explain it in the results. Why, if PPY alone is so nontoxic, does it cause 35% of cells to die? What happens to control cells with nothing administered? 100% viability? Please clarify.
Figure 6. Are serum ALT and AST levels not significantly different than saline when comparing LPB+DOX + PPY PTT and Saline? I assume so, please state that. (Line 418, watch your treatment of significant figures, 54 +/- 14, 28 +/- 4 etc). Figure 6 Axis labels are illegible. Make a legend key on the side or define abbreviations in the Figure legend. Don’t make the reader zoom in to 250% to get results. Also, make the image sharper, clearer, and use larger fonts for Y axis (you have space).
Author Response
Dear reviewer:
Thank you for your comments concerning our manuscript and offering us opportunity to improve the quality of our submitted manuscript entitled “Combined photothermal therapy and Lycium barbarum polysaccharide for topical administration to improve the efficacy of doxorubicin in the treatment of breast cancer”. Those comments are valuable and very helpful. We have read through comments carefully and have made corrections. Based on the instructions provided in your letter, we uploaded the file of the revised manuscript. We submit here the revised manuscript as well as a list of changes. The responses to the reviewer's comments are marked in red and presented following. Our point-to-point responses to the queries raised by the reviewers are listed, please see the attachment. We sincerely hope this manuscript will be finally acceptable to be published on Pharmaceutics: Nanomedicine and Nanotechnology (Polymer Nanoparticles for the Delivery of Anticancer Drugs). Thank you very much for all your help and looking forward to hearing from you soon. Sincerely wish you health, happiness and all the best!
Best regards
Sincerely yours
Meihua Han and Lina Sun

Round 2
Reviewer 1 Report (New Reviewer)
Suggestions after second revision
In the introduction they mention "Among the existing treatments for breast cancer, including radiotherapy and chemotherapy", there are also drugs based on hormones, monoclonal antibodies, aptamers and nanoparticles without the need for photothermal therapy. Mention it.
Studies have shown that anthracyclines can significantly reduce breast cancer recurrence rates and reduce mortality [7]. Epidemiological data are missing, as is the geographical delimitation (regions or continent). add it.
Line 57 should only mention L. barbarum, please check in text after the first mention in the text.
In scheme1 I consider that if it has value as the main figure of the article if they are added, a hypothetical scheme of interaction at the molecular level could be made within the tumor cell with the treatment.
The images in Fig. 1 lack specifications of the figure, dimensions, sizes, the name of the techniques used.
Author Response
Dear reviewer:
Thank you for your comments concerning our manuscript and offering us opportunity to improve the quality of our submitted manuscript entitled “Combined photothermal therapy and Lycium barbarum polysaccharide for topical administration to improve the efficacy of doxorubicin in the treatment of breast cancer”. Those comments are valuable and very helpful. We have read through comments carefully and have made corrections. Based on the instructions provided in your letter, we uploaded the file of the revised manuscript. We submit here the revised manuscript as well as a list of changes. The responses to the reviewer's comments are marked in red and presented following. Our point-to-point responses to the queries raised by the reviewers are listed, please see the attachment. We sincerely hope this manuscript will be finally acceptable to be published on Pharmaceutics: Nanomedicine and Nanotechnology (Polymer Nanoparticles for the Delivery of Anticancer Drugs). Thank you very much for all your help and looking forward to hearing from you soon. Sincerely wish you health, happiness and all the best!
此致敬意
真诚的你
韩梅华、孙丽娜

This manuscript is a resubmission of an earlier submission. The following is a list of the peer review reports and author responses from that submission.
Round 1
Reviewer 1 Report
This work describes the use of a therapy that combines the LPB polysaccharide, plus photothermal therapy under NIR irradiation and pyrrolidone nanoparticles, plus gelatine, as site-specific treatment against a xenograft breast tumour modelled in mice.
The manuscript needs to be improved in depth, since it is very difficult to be followed; after that, it will be possible to truly appreciate the value of the work.
Just some observations to exemplify how confuse the text is:
0) The English language of the entire manuscript must be improved.
1) The sentences in the abstract are too long; that makes difficult their comprehension: 4 rows without a point are not fair...
2) Rows 62-66: the authors state that the local administration avoids the contact with healthy organs, but the stealth liposomal doxorubicin already possesses such ability. It is unclear hence, which are the differences between such approach and the one the authors propose (or its advantages). Please explain better your point. Is the palmar planta toxicity from stealth liposomal doxo, which is systemically administered?
3) Please explain better the effects/mode of action of LBP. How can the LBP reduce the mitochondrial membrane potential and being non-cytotoxic at the same time?
4) Scheme 1: the current size is too small.
5) Materials section is uncomplete. The Poloxam 188 and the polymer polypyrrole (PPY) origins are lacking.
6) Section 2.3. “Preparation of drug-containing gelatine” is completely incomprehensible: please re-write it.
7) Section: 2.4.1. “Preparation of PPY NPs”: is the described method original or is it based in a previously developed method? If that was the case, please include the reference.
8) “Investigation of photothermal conversion performance” section : was the determination of T every 30 s aimed to adjust the concentration of nanoparticles? What is the relationship between T and nanoparticles concentration, if the aim of the measure was to determine the photothermal conversion performance?
9) 2.7. “In vivo tissue distribution study” section : The nature of the gelatine injection is confuse: are the gelatine macro, micro or nanoparticles? On the other side, 100 mm3 corresponds to small spheric tumours of nearly 2,9 mm radius, small for humans but huge for mice. Such tumours must be nearly on the skin surface, to be identified and irradiated, and not deep enough in the tissues, as occurs in clinical settings. That raises me doubts about the validity of the model…
10) Rows 198-199 and 207-211: the dosage form is difficult to understand. For instance, which is the difference between: (2) 2 mg/kg DOX (iv) y (4) 2 mg/kg DOX solution?. I suggest the use of a table in order to describe better the formulations and administration route received by the 13 groups.
11) Rows 212-213: please re-write the sentence to make it clearer
12) Rows 217: TSR, CIR, LIR, SIR, LuIR, RIR, and TIR. Please indicate immediately their meaning and not after.
Reviewer 2 Report
The authors has studied the photothermal local delivery of the drug combination ( Lycium barbarum polysaccharide and DOX) to the tumor. This is the routine drug delivery research with no novelty involved. Combination of Lycium barbarum polysaccharide and DOX was already reported and author failed to provide the novelty of the present research work .
In my opinion the paper has several shortcomings which is not just limited to the English language but I found huge gaps in the continuity of the work flow and data analysis. There are several areas in the MS that deserve improvements. AT this point, before even going into the technical aspect of the research I suggest the improvement in the English language. This is very badly written manuscript and hence Authors should take professional help to improve the communication . Additionally, better citations will help the authors to provide better and more accurate information regarding research work they conducted. My specific observations are given below.
Comment 1: The title needs improvement, it’s an external stimulus(NIR) mediated release of the drug combination, however title is focused only on the local delivery of the DOX.
Comment 2: Sentence between line 21 to 28 is too long to comprehend. Please divide it into at least 2 sentences. Please take professional help to improve English.
Line 28 to 31: The results of biochemical assays showed a statistically significant 28reduction in ALT and AST levels in the combined topical administration group compared to the DOX alone group (ALT 54 vs. 28; AST 158 vs. 111) (P<0.05), indicating that liver damage was sig-30nificantly lower in the combined topical administration group.
Possible replacement for the above sentence: The results of biochemical assays showed a statistically significant reduction in ALT and AST (ALT 54 vs. 28; AST 158 vs. 111) (P<0.05), indicating that liver damage was significantly lower in the combined topical administration group.
Comment 3: Line 30: (ALT 54 vs. 28; AST 158 vs. 111) SD values are required.
Comment 4: Line 33: mention anti-inflammatory markers studied (inhibited ) using Elisa assay.
Comment 5: Line 42 to 46:
First line is the reference to the statistics while “malignant lesions occur in the catheter behind the areola in men” is the pathological manifestation. Please divide this information in different sentences.
Comment 6: Line 64: drug distribution at the tumor site while reducing the drug accumulation at non-tumor site: Change to “drug distribution in the tumor while reducing…………….
Comment 7: Line 66: Therefore, on the basis of local administration, we explore local administration methods that can improve the efficacy of doxorubicin in the treatment of breast cancer, hoping to improve the therapeutic effect and reduce the toxic 68side effects of doxorubicin in the treatment through multi-faceted synergistic treatment.
Rewrite as Therefore, on this bases, in the present investigation, we explored local administration methods to improve the efficacy of doxorubicin in breast cancer treatment, hoping to improve the therapeutic effect and reduce the toxic side effects of doxorubicin in the treatment through multi-faceted synergistic treatment.
Comment 8: Line 73-73: Incorrect : rewrite it.
Comment 9: Line 74 to 76:
In terms of anti-tumor, it has been demonstrated that the main active component of LBP (molecular weight 40-350 kDa) can inhibit the growth of H22 cells in vitro, induce apoptosis in H22 cells, make the mitochondrial membrane potential disruption, cause S-phase block, and have no significant toxicity to mice in vivo.
Rewrite it as “In terms of anti-tumor activity , it has been demonstrated that the main active component of LBP (molecular weight 40-350 kDa) can inhibit the growth of H22 cells in vitro, induce apoptosis, disturb mitochondrial membrane potential, cause S-phase block, and have no significant toxicity to mice in vivo”
Comment 10: Line 85-91: This is one single sentence. It’s difficult to comprehend. Divide information in at least 2- 3 sentences
Comment 11: Please rewrite whole introduction as most of the sentences are grammatically incorrect or else difficult to comprehend.
Comment 12: Line 129: Preparation and characterization of nanoparticles?
Comment 13: Section :2.4.1. This section is written incorrectly. Please recheck whole manuscript for grammar and typos.
Comment 14: Line 144: The PPY NPs samples were measured three times in parallel. Measured in parallel for what ?
Comment 15: Line 144: sentence is starting with And. its incorrect. rewrite whole paragraph.
Comment 16: Line 148: 8% NaCl, 10% glucose and PBS,
Is this composition of the physiological medium ?
Comment 17: Line 141 ,147, 151. Give subheading number.
Comment 18: Line 152 to 152: 1mLPPYNPs was added to the cuvette and a temperature sensor was inserted. After preparation and stabilization, the solution temperature recorded was the solution temperature at 0s.
Rewrite it.
Comment 19: Line 165: certain concentration gradient
Which gradient the author is talking about?
Comment 20: 182: How many mice
Comment 21: Section 2.7. in vivo tissue distribution study and Section 2.8. In vivo antitumore study :
For each section, Please provide the tabulated summery of the different treatment groups, no of animals in each group , concentration of each compound, type of the treatment and formulation, NIR treatment , NIR intensity and wavelength.
Comment 22: Line 215: Avoid use of the word “we”
Comment 23: Section 3.1. 3.1. Preparation of drug-containing gelatin
Rewrite this section. The word “after” in two consecutive sentences without prior context is incorrect .
Comment 24: Line 256:
The particle size characteristics of nanoparticles are favorable for pharmacokinetics and cellular uptake in vivo
Should be written as “The particle size of nanoparticles are favorable for pharmacokinetics 256and cellular uptake in vivo”
Comment 25: Line 257: As shown in Figure 1B and C, the average particle size of PPY NPs was 229.30 nm, respectively.
What respectively is referring here?
Comment 26: Section : 3.2. Preparation and characterization of PPY NPs.
Provide span values.
Comment 27: Line 262 : does NP improves solubility ? Line 260 and 261 is incorrect.
Comment 28: Line 266 to 269 is written incorrectly .
Comment 29: Section : 3.5. In vitro cellular uptake
This section contains only one sentence which is difficult to understand.
Comment 30: Line 391: Does the high temperature increases drug penetration and drug endocytosis in cells ? does author want to say that drug endocytosis is temperature depended?
Comment 31: Line 337: biocompatible photothermal conversion material PPY NPs can improve the absorption efficiency of 338NIR by tumor tissues,
How PPY improves the absorption? Or does it improves the drug release when irradiated with NIR ?
Comment 32: Line 377 to 387 is one sentence. It’s impossible to read and understand.
Comment 33: 3.9. Serological analysis.
Rewrite second paragraph of this section. If possible, before resubmission get it checked from the professional.
Reviewer 3 Report
The manuscript presents the results of large and diversified experiments focusing effect of a complex therapeutic combination ( of Lycium barbarum polysaccharide (LBP) extract, and doxorubicin (DOX)into a nano-polypyrrole nanoparticles (PPY) construct further combined with photothermal irradiation. The results of in vitro and in vivo are adequately presented and highlight the effect of the different moieties and their combination. All the components both cytotoxic agents and therapeutic showed their effect in these experiments: since the effect of PPY Nps was already reported
in Biomacromolecules 2019 (doi.org/10.1021/acs.biomac.8b01453), the main weakness of the manuscript is to identify a synergistic effect between the different therapeutic options and better justify the statement (working hypothesis) about the dose of DOX reduction administration since the NPs did not present a targeting capacity, or this aspect has not been evidenced.
Notwithstanding, the comparison between the performances of a pure PTT therapy, as reported in Biomacromolecules (2019) and the combined effect (in this case the combination of herbal and well know cytotoxic agents rises higher antitumour activity?). Also, the performances of the NPs loaded in terms of PT effect are not clearly evidenced in the manuscript. Focusing on the bioavailability issue, it’s not clear the PK of the loaded PPY NPs and the extraction in the tumours, nor the local distribution of the NPs in the lesion when the administration is performed locally. The simplified approach using PTT and PPY NPS in cell culture is not presented, although is feasible this has been done before the in vivo experiments: are the performances of the loaded PPY NPs clearly enhanced after the drugs loading under laser irradiation?
According to these aspects, the manuscript will benefit from a revision for the better elaboration of the mechanistic effects and highlighting the performances of the PPY platform for local ( and systemic ?) drug delivery and therapy.